# Development of an ELISA-Based Potency Assay for Inactivated Influenza Vaccines Using Cross-Reactive Nanobodies

**DOI:** 10.3390/vaccines10091473

**Published:** 2022-09-05

**Authors:** Chung Y. Cheung, Sitara Dubey, Martina Hadrovic, Christina R. Ball, Walter Ramage, Jacqueline U. McDonald, Ruth Harvey, Simon E. Hufton, Othmar G. Engelhardt

**Affiliations:** 1Vaccines Division, Scientific Research & Innovation, Medicines and Healthcare Products Regulatory Agency, South Mimms, Potters Bar EN6 3QG, UK; 2Biotherapeutics Division, Scientific Research & Innovation, Medicines and Healthcare Products Regulatory Agency, South Mimms, Potters Bar EN6 3QG, UK

**Keywords:** influenza, vaccine, potency test, nanobodies, ELISA, cross-reactive

## Abstract

Inactivated vaccines are the main influenza vaccines used today; these are usually presented as split (detergent-disrupted) or subunit vaccines, while whole-virus-inactivated influenza vaccines are rare. The single radial immune diffusion (SRD) assay has been used as the gold standard potency assay for inactivated influenza vaccines for decades; however, more recently, various alternative potency assays have been proposed. A new potency test should be able to measure the amount of functional antigen in the vaccine, which in the case of influenza vaccines is the haemagglutinin (HA) protein. Potency tests should also be able to detect the loss of potency caused by changes to the structural and functional integrity of HA. To detect such changes, most alternative potency tests proposed to date use antibodies that react with native HA. Due to the frequent changes in influenza vaccine composition, antibodies may need to be updated in line with changes in vaccine viruses. We have developed two ELISA-based potency assays for group 1 influenza A viruses using cross-reactive nanobodies. The nanobodies detect influenza viruses of subtype H1N1 spanning more than three decades, as well as H5N1 viruses, in ELISA. We found that the new ELISA potency assays are sensitive to the nature of the reference antigen (standard) used to quantify vaccine antigens; using standards matched in their presentation to the vaccine type improved correspondence between the ELISA and SRD assays.

## 1. Introduction

Most influenza vaccines are trivalent or quadrivalent inactivated vaccines, containing three or four different influenza virus strains that are produced either in embryonated hens’ eggs or in cell culture. These are reformulated regularly to keep pace with virus evolution, following the biannual recommendations of the World Health Organisation (WHO) Global Influenza Surveillance and Response System (https://www.who.int/teams/global-influenza-programme/vaccines/who-recommendations (accessed on 21 July 2022)). The potency of the inactivated influenza vaccines is determined by measuring the content of haemagglutinin (HA) antigen using the well-established single radial immune diffusion (SRD, also known as SRID) assay [1,2,3,4]. The SRD assay requires two types of reagents, which are supplied to vaccine manufacturers and independent control laboratories by the four WHO Essential Regulatory Laboratories: a calibrated antigen reference reagent and a sheep antiserum specific for one of the virus components of the vaccine. Both of these reagents are updated every time a new vaccine strain is included in the vaccine. The production of sheep antisera can take six to more than ten weeks and is one of the rate-limiting steps in the calibration of new antigen reference reagents, influenza vaccine potency testing, and vaccine release.

The search for alternative potency assays for inactivated influenza vaccines has intensified since the 2009 influenza pandemic [5] and has resulted in a range of proposed assays [1,4,6,7,8,9,10,11,12,13,14]. It is widely accepted that a new potency assay should be able to distinguish native HA antigens from denatured or structurally altered antigens. The use of immunological reagents (such as polyclonal or monoclonal antibodies) is likely to enable such discrimination. However, many antibodies to influenza HA are narrowly specific for the influenza virus strain they are raised against and closely related viruses, necessitating the generation of new antibodies in response to a strain change in a seasonal influenza vaccine or when a new pandemic influenza virus emerges, which entails limits on the timeliness of implementation of potency assays in these scenarios.

In recent years, monoclonal antibodies targeting the conserved stem or stalk domain of the HA have been described [15,16,17,18,19]. These antibodies are more cross-reactive than most antibodies binding the globular head domain of HA, in some cases permitting the recognition and neutralisation of viruses belonging to different subtypes, especially those subtypes belonging to the same HA phylogenetic group (groups 1 and 2). We recently developed a panel of single-domain antibodies (nanobodies) that are cross-reactive within phylogenetic group 1 [20].

Here, we report the development of two new ELISA-based potency assays for inactivated influenza vaccines that make use of cross-reactive nanobodies.

## 2. Materials and Methods

### 2.1. Nanobodies, Reference Reagents and Vaccines

Nanobodies R1a-B6, R2b-D9, R2b-E8, R1a-A5, and R2a-G8 were expressed and purified as previously described [20]. Briefly, nanobodies cloned in the pNIBS-1 phage display vector [20] were transformed into Escherichia coli WK6 cells. The cells were grown at 37 °C in 2 × YT supplemented with carbenicillin (100 μg/mL) and 0.1% (*w*/*v*) glucose to an OD600 nm 0.6–0.8. Expression was induced by the addition of IPTG to a 1 mM final concentration followed by incubation overnight at 28 °C. The periplasmic extracts were prepared using osmotic shock [20] and his-tagged nanobodies purified by immobilised metal chelate chromatography (IMAC) using TALON™ resin (Clontech, Takara Bio Inc., Kusatsu, Japan), according to manufacturer’s instructions. The purified samples were dialysed into PBS using Slide-A-Lyzer^®^ cassettes with a 3.5 kDa molecular weight cut-off (Pierce, Thermo Fisher Scientific, Waltham, MA, USA). The nanobodies were further purified using size-exclusion chromatography on a Superdex™ 75 10/300 GL column (GE Healthcare, Chicago, IL, USA) run in PBS. The pooled fractions were sterile-filtered using millex^®^–GV filter units (Millex), and the size and the purity were assessed by analytical SE-HPLC and SDS-PAGE. For the sandwich ELISA, the nanobodies were re-cloned into pNIBS-1 with the c-Myc epitope tag removed and expressed and purified as above. The nanobodies without a c-Myc tag were biotinylated using an EZ-Link^®^ Sulpho-NHS-SS-Biotinylation kit (Pierce, Thermo Fisher Scientific) and the molar ratios of biotin to nanobody were determined using a biotin quantitation kit (Pierce, Thermo Fisher Scientific). For in vivo biotinylation, the nanobodies without a c-Myc detection tag were cloned into the pAviTag C-His kan vector transformed in Biotin XCell F’ cells. The cells were grown in LB containing kanamycin (30 μg/mL) to an OD600 nm of 0.6–0.8 followed by the induction with 0.2% (*w*/*v*) rhamnose, 0.01% (*w*/*v*) arabinose and 50 μM biotin at 30 °C overnight (Lucigen, Middleton, MI, USA). The expressed nanobody was released from the periplasm by osmotic shock and purified as above.

The list of reference reagents used is shown in Table 1. All of the reference reagents were reconstituted in deionized water according to their instructions for use (NIBSC).

The split vaccine was obtained from two manufacturers: 1 batch of trivalent split vaccine (V1) from manufacturer A; 1 batch of H1N1 monovalent split vaccine (V2), 1 batch of quadrivalent split vaccine (V3), 2 batches of trivalent split vaccine (V4 and V5) from manufacturer B. The H1N1 component in all of the vaccine samples was A/California/7/2009 NYMC X-181.

### 2.2. SRD Assay

The SRD assay was carried out according to the European Pharmacopoeia and as described previously [21]. Briefly, antiserum was included in an agar matrix into which wells were cut. The vaccine samples or the antigen reference reagents added to the wells were allowed to diffuse into the matrix and react with the antiserum for a minimum of 18 h at 20–25 °C. The precipitin zones were measured following staining with Coomassie Blue.

### 2.3. Competitive ELISA

The antigen used to coat the 96-well plates (F96 Maxisorp NUNC-immuno plates, Thermo Fisher Scientific) for the competitive ELISA was the A/California/7/2009 (NIBRG-121xp) egg-derived antigen reference reagent NIBSC 09/196. A 1:50 dilution was made in phosphate-buffered saline (PBS), and 100 µL was added to each well so that each well contained 74 ng of the antigen. The plates were then incubated overnight at 4 °C, after which the plates were blocked for 1 h with 2% skimmed milk (Marvel, Premier Foods Group, St. Albans, UK) in PBS; 2-fold dilutions of the antigen reagents and the vaccine samples were made on the plates so that eight dilutions were tested in duplicates (minimum number of replicates) or more. The primary antibody in PBS with 2% skimmed milk was added at 0.25 µg/mL and then incubated on a rotary shaker (PHMP-4, Grant-bio) at 37 °C for 1 h. After diluting the secondary antibody against c-Myc (ab19312, Abcam, Cambridge, UK) at 1:5000 with 2% skimmed milk in PBS, 100 µL was added to each well. After incubation at 37 °C for 1 h on a rotary shaker, 100 µL of 3,3’,5,5’-Tetramethylbenzidine (TMB) (Europa Bioproducts, Ipswich, UK) was added, and the reaction was stopped by the addition of 50 µL 1 M sulphuric acid after 15 min of incubation at room temperature. The absorbance was obtained by reading the plates at 450 nm with a spectrophotometer (Spectra Max M3, Molecular Devices, San Jose, CA, USA) and data generated with Softmax Pro (Molecular Devices).

### 2.4. Sandwich ELISA

Biotinylated R1a-B6 (c-Myc negative version) nanobody was used as the coating/capture antibody, which was immobilised on pre-blocked streptavidin-coated 96-well plates (Thermo Fisher Scientific) at 0.3 μg/mL by overnight incubation in PBS at 4 °C. The reference antigens or test samples were treated with Zwittergent 3–14 detergent (Calbiochem-Behring, La Jolla, CA, USA) for 30 min at room temperature. Two-fold serial dilutions of the antigen samples were made in separate plates, and the dilutions were added to the coated plates in the presence of 1% *w*/*v* Zwittergent 3–14 in duplicate (minimum number of replicates) or more and incubated at 25 °C on a rotary shaker for 1 h. R2b-D9 at 0.3 μg/mL was used as the primary detection antibody and incubated at 25 °C on a rotary shaker for 1 h. An antibody against c-Myc (ab19312, Abcam) was used as secondary detection antibody at 1:5000 dilution, which was incubated at 25 °C on a rotary shaker for 1 h. TMB was used as an enzyme substrate, and after incubation for 15 min at room temperature, the reaction was stopped with 1M sulphuric acid and the plates were read on a spectrophotometer (Spectra Max M3, Molecular Devices) at 450 nm and data generated with Softmax Pro (Molecular Devices).

### 2.5. Calculation of Estimated Potency

The ELISA data were analysed with CombiStats software (EDQM, Council of Europe, Strasbourg, France) to determine the estimated potency. Parallel line analysis used four parameter regression fitting of test samples compared to a reference antigen standard. The replicate test samples were included on each plate, and the replicate plates were included in each assay.

### 2.6. Forced Degradation

For the forced degradation by heat, the reference antigen reagent NIBSC 09/196 was reconstituted according to the instructions and incubated in a water bath at 56 °C for 15 min, 48 h, and 64 h. These treated samples were stored at 4 °C and assayed the next day by SRD and the two different ELISA formats along with the untreated antigen reagent.

For the forced degradation by deamidation, the antigen reagent NIBSC 09/196 was reconstituted according to the instructions and subjected to treatment with 40 mM N-cyclinhexyl-3-aminopropanesulfonic acid (CAPS) at pH 11.0 at 32 °C for 5, 15, and 60 min to induce deamidation. Then, 500 mM citrate at pH 3.0 was added to a final concentration of 2% (*v*/*v*) to stop the reaction. The samples were stored at 4 °C until analysis on the next day by SRD and both ELISA formats.

For the forced degradation by low pH, the antigen reagent NIBSC 09/196 was reconstituted according to the instructions and subjected to treatment at pH 4 at room temperature for 30 min by the addition of 2M HCl. Neutralisation was achieved by the addition of 1M NaOH until the pH reached 7.0 while monitoring the pH throughout using the Pocket Checker1 pH Tester (Hanna, Leighton Buzzard, UK). The samples were then stored at 4 °C until use on the next day in SRD and ELISA.

## 3. Results

### 3.1. Cross-Reactivity of Nanobodies in a Competitive ELISA Format

We tested five nanobodies with known H1–H5 cross-reactivity [20] in a competitive ELISA format. An antigen reagent made from virus A/California/7/2009 NIBRG-121xp (NIBSC code 09/196) was used as coating antigen. Antigen reagents representing the H1N1pdm09, pre-2009 H1N1, and H5N1 viruses (Table 1) were used as competing antigens, with reagent 09/196 used as standard. The values estimated for all antigens by parallel line analysis based on the 09/196 standard were compared to the values of these antigen reagents previously assigned using SRD. The results are shown in difference plots in which the percent difference between the estimated and assigned potency for each reagent is plotted against the assigned potency; the average deviation of the measured values from the assigned potencies is shown as a red dotted line (Figure 1). All five nanobodies reacted well with the H1N1pdm09 antigens (Figure 1A). The average deviation of the estimated potency values from the assigned values ranged from −22.35% to 38.48% for the five nanobodies. When we tested antigens covering H1N1 viruses from 1976 to 2007, four of the nanobodies recognised all of the antigens well, whereas nanobody R2b-E8 only reacted with one antigen, A/New Jersey/8/76 (code 77/530) (Figure 1B,C). The average deviation of the estimated values from the assigned values was from −66.65% to 49.73% (Figure 1B). All of the nanobodies reacted with the H5N1 antigens tested, with an average deviation of the estimated values from the assigned values from −54.87% to 0.13% (Figure 1D). We also assessed the correlation between estimated and assigned values for all nanobodies by antigen groups (H1N1pdm09, H1N1, H5N1) (Table 1). Pearson’s correlation coefficients were best for viruses belonging to subtype H1N1pdm09 (>0.92 in all cases), lower for pre-2009 H1N1 viruses (0.712–0.879), and lowest for H5N1 viruses (0.434–0.707) (Table 2). These findings suggest that the discrepancies seen when comparing estimated with assigned values (Figure 1) were related to the standard used rather than due to intrinsic properties of the nanobodies or the assay. The same H1N1pdm09 antigen reagent, derived from A/California/7/2009 NIBRG-121xp, was used as standard in all assays reported in Figure 1; as viruses’ antigenic difference increased from the standard (H1N1pdm09 vs. pre-2009 H1N1 vs. H5N1), the correlation decreased. With different standards matched to the antigen being measured or re-calibrated standards, a better agreement between the estimated values and those assigned by SRD may be achievable.

For further work with the competitive ELISA, nanobody R2b-D9 was chosen.

### 3.2. Sandwich ELISA

For the development of a sandwich ELISA, we selected nanobody R1a-B6 as the capture antibody and R2b-D9 as detection antibody. Nanobody R1a-B6 was reformatted to remove the c-Myc tag and was biotinylated in vitro for efficient coating of streptavidin plates. In the initial experiments, the signal was low when A/California/7/2009 NIBRG-121xp antigen 09/196 was captured on the R1a-B6-coated plates and detected using R2b-D9 and a secondary anti-myc antibody (data not shown). As the SRD assay uses Zwittergent 3–14 detergent to pre-treat the antigen or vaccine samples, we treated the antigen with 1% Zwittergent 3–14 in PBS, the same concentration of detergent used in the SRD assay, before dilution and detection in the sandwich ELISA. The signal substantially improved when Zwittergent was used (Figure 2A). Thus, we used Zwittergent in all subsequent sandwich ELISA experiments.

Antigen reagents representing the H1N1pdm09, pre-2009 H1N1, and H5N1 viruses (Table 1) were measured in the sandwich ELISA using antigen reagent 09/196 as standard. The results are shown in Figure 2B–D as difference plots. All of the antigens were recognised by the nanobodies employed in the sandwich ELISA. Average deviations of estimated values from assigned value were 3.83% for pre-2009 H1N1 (Figure 2B), 15.52% for H1N1pdm09 (Figure 2C) and 32.52% for H5N1 viruses (Figure 2D).

### 3.3. The ELISA Potency Assays Are Stability-Indicating

To assess whether our competitive and sandwich ELISAs were able to distinguish the native from denatured HA, we conducted forced degradation experiments that were similar to those conducted previously for other potential influenza vaccine potency assays [6,22,23]. We subjected antigen 09/196 to heat treatment, deamidation, and acid treatment and tested the treated and untreated samples in the SRD assay and both ELISA formats. Antigen was undetectable by SRD assay after treatment at 56 °C for 15 min or longer (Figure 3(Ai)). The potency values decreased by 70% and 87% after 15 min exposure to a temperature of 56 °C when measured by competitive and sandwich ELISA, respectively (Figure 3B). Longer heat treatment resulted in undetectable antigens in both ELISA assays (Figure 3B).

Deamidation with CAPS for 5 min destroyed all reactivity of the antigen in the SRD assay (Figure 3(Aii)) and led to decreases of 70% and 90% in the competitive and sandwich ELISA, respectively (Figure 3B). Longer deamidation eliminated all of the signals in the ELISA assays. After the antigen had been exposed to a pH of 4 for 30 min, none of the three assays detected any antigen (Figure 3(Aiii) and data not shown).

### 3.4. Selectivity

We developed the two ELISA assays deliberately for wider cross-reactivity than the currently used SRD assay; however, it is important that the different components of multivalent influenza vaccines can be measured independently of each other. H3N2 viruses, which are the second influenza A virus component in trivalent and quadrivalent vaccines, belong to phylogenetic group 2, while influenza B viruses are not expected to react with antibodies raised against type A viruses. We tested three antigen reagents, one for H1N1, one for H3N2 and one for influenza B, in the two ELISA formats (Table 1). As is shown in Figure 4, the ELISA assays displayed appropriate selectivity, with no response observed for H3N2 and B antigens. When all three antigens were mixed, reflecting the situation of a trivalent influenza vaccine, the response curves closely mirrored the ones obtained with H1N1 antigen only, suggesting that no interference occurred (compare red and black lines in Figure 4).

### 3.5. Linearity and Limit of Detection

Using two-fold dilutions of antigen 09/196, we determined the lower limit of detection as a dilution of 1/8, corresponding to 4.6 µg/mL of HA, for the competitive ELISA (Figure 5A). Linear regression showed r^2^ = 0.9935 and the y-intercept from the equation at −1.5189.

The sandwich ELISA had a lower limit of detection at a dilution of 1/32, corresponding to 1.2 µg/mL HA. Linear regression analysis gave r^2^ = 0.9999 and a y-intercept at 0.1992 (Figure 5B).

### 3.6. Testing of Vaccine Samples

All of the experiments reported so far were conducted with antigen reagents, which are whole virus preparations. Seasonal influenza vaccines, however, are usually further processed into detergent-disrupted (split) virion or subunit vaccines. To assess whether the ELISA assays would work with vaccine material, we tested split vaccines from two manufacturers (Table 3 and Table 4). All of the products contained the same H1N1 strain, A/California/7/2009 NYMC X-181 (H1N1)pdm09. We determined a reference value for each vaccine by SRD assay; for ELISA, we used the same homologous antigen reagent, 12/168, as the standard that was used in the SRD assay. The values obtained by ELISA differed from the SRD reference values when using antigen reagent 12/168 as standard by 64 to 142% in the competitive ELISA (Table 3) and from 59 to 91% in the sandwich ELISA (Table 4). We hypothesised that the nature of the antigen being measured, i.e., whole-virus in the case of the standard and split virus in the case of the vaccine, might affect the values determined. Therefore, we recalculated the results using each of the vaccine samples in turn as a standard, with its SRD value assigned as its calibration value, for all other vaccine samples. This procedure reduced the difference between the potencies measured by SRD and those measured by ELISA: differences were −23 to +42% for the competitive ELISA (Table 3) and −12 to +24% for the sandwich ELISA (Table 4), indicating that the use of a standard in a presentation equivalent to the tested vaccine sample improves the agreement between SRD and ELISA.

## 4. Discussion

Potency testing of vaccines is a crucial component of vaccine quality control; from the formulation of vaccine to the final lot release by independent control laboratories, a potency test is used to ensure that the correct amount of antigen/immunogen is present in the vaccine. For inactivated influenza vaccines, the SRD assay has been in use as the gold standard potency test since the late 1970s [24]. Recently, interest has grown in the development of alternative potency assays to address some of the perceived shortcomings of the SRD assay, such as the time needed to prepare annually updated antiserum reagents, the unsuitability of the SRD assay for high-throughput formats and the relatively low sensitivity, which may be relevant for future low-dose adjuvanted (pandemic) vaccines [5]. While some physicochemical assays promise speed and good sensitivity, most of these are not stability-indicating, i.e., they cannot distinguish between native, immunologically active, and denatured HA antigens [22,23], although recent developments may allow antibody-free detection of native HA [25,26]. Immunological assays are more likely to be suitable alternatives to the SRD assay. However, because such assays usually require antibody reagents, they may suffer the same problem that the SRD assay encounters, namely the need for regularly updated antibody reagents, the generation of which is time-consuming and may lead to delays in vaccine formulation and release.

We, therefore, attempted to circumvent this issue by using nanobodies selected for broad reactivity with HA proteins of several subtypes belonging to phylogenetic group 1 [20]. Nanobodies were employed in two ELISA formats, a competitive and a sandwich ELISA format. Both formats were shown to work with multiple strains of influenza virus. Importantly, our ELISA assays were stability-indicating and showed good linearity and reasonable limits of detection, with the sandwich ELISA proving more sensitive.

With the use of broadly reactive nanobodies, we have solved one rate-limiting step of the SRD assay: these nanobodies are expected to react with future group 1 HA antigens and, therefore, new antibody reagents will not have to be generated for every strain change in the influenza vaccine. However, it would be prudent to monitor the performance of our nanobodies with future H1N1pdm09 seasonal influenza viruses. We have previously shown that escape mutants with amino acid substitutions in the HA stalk region that react poorly with these cross-reactive nanobodies can be generated [27]. As we know amino acid positions that are important for binding the nanobodies used in the presented ELISA assays, it should even be possible to predict, from sequence analysis alone, whether a new influenza virus strain is likely to lose binding to the nanobodies. In such cases, and if binding were shown to be reduced experimentally, it would be possible to use other cross-reactive nanobodies with partially overlapping epitopes [20,27] or to derive new nanobodies through the selection of nanobodies against the HA of the new virus from existing phage display libraries or through in vitro affinity maturation/evolution of the nanobodies currently used in the two ELISA formats. Processes to generate new nanobodies would need to be optimised to ensure a timely supply of replacement binding reagents in case of the loss of reactivity of the existing nanobodies.

Apart from antibody reagents, immunological potency assays such as the SRD assay or ELISA assays require calibrated antigen reference reagents. For the SRD assay, homologous antigen reagents are used: the antigen reagent (standard) for measuring a vaccine component is derived from the same candidate vaccine virus as the one used in vaccine production. Thus, antigen reagents are also changed every time the vaccine composition changes, and in many seasons, more than one antigen reagent is required due to the use of several candidate vaccine viruses by different manufacturers. Here we used a non-homologous antigen reagent in most assays (reagent 09/196) and found that values obtained by ELISA did not exactly match those assigned by the SRD assay (Figure 1 and Figure 2). In general, the further the antigenic/genetic distance of the test virus was from the virus contained in the standard, the poorer the agreement between SRD and ELISA and the lower the correlation between the potency values obtained by the two assays. This may reflect subtle differences in the binding of different HA molecules to the nanobodies, even though the structure of the stalk region is more conserved than the head domain. When we tested vaccine samples, we used a homologous antigen standard but again found discrepancies between the ELISA and the SRD assay. We hypothesise that the different presentations of the antigen, either in whole virions (standard) or in micelles (vaccines), affected the outcome. Indeed, when we used vaccine samples as standards, agreement with the SRD improved, suggesting a way towards better antigen standards for the ELISA: split virus standards may be better for determining the potency of split vaccines by ELISA than SRD standards, which are whole-virus preparations. Similarly, subunit or recombinant HA standards may be appropriate standards for subunit and recombinant vaccines, respectively.

In conclusion, we report here the development of a potency assay based on ELISA using cross-reactive nanobodies for group 1 HA. Our two ELISA assay formats could be further developed, including for the other vaccine components of trivalent or quadrivalent influenza vaccines, with the appropriate nanobodies.

## Figures and Tables

**Figure 1 vaccines-10-01473-f001:**
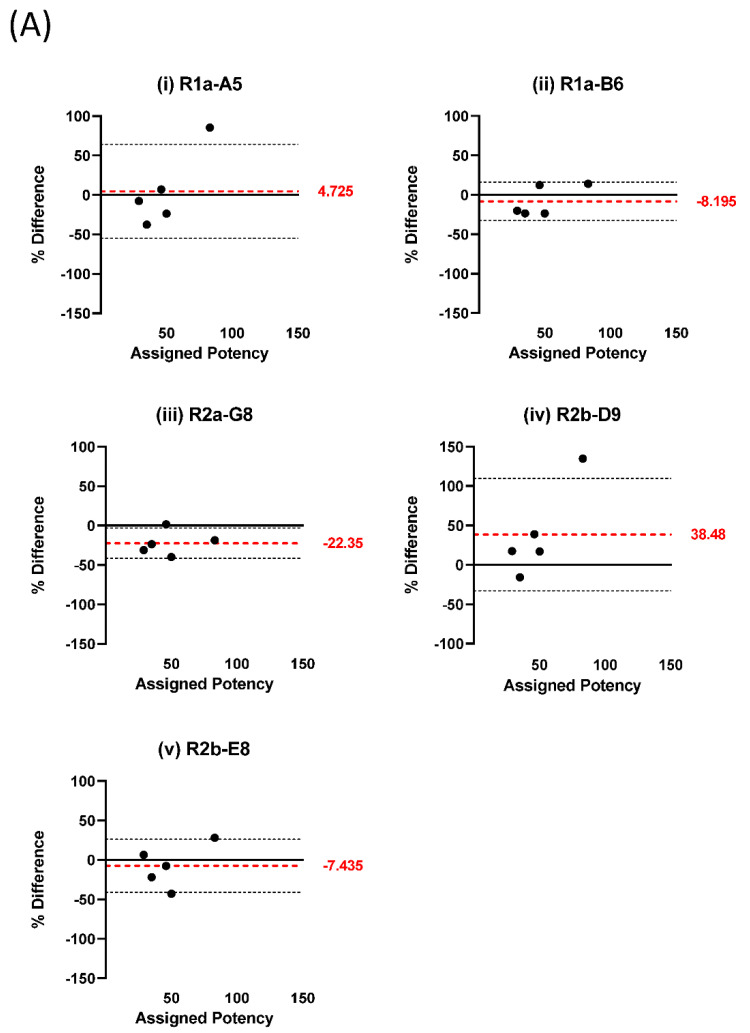
Difference plot analysis of results from competitive ELISA using different nanobodies. The percent differences between assigned values (in μg/mL) and values estimated by competitive ELISA with nanobodies R1a-A5 (**i**), R1a-B6 (**ii**), R2a-G8 (**iii**), R2b-D9 (**iv**) and R2b-E8 (**v**) are shown for (**A**) A(H1N1)pdm09 antigen reagents, (**B**) A(H1N1) antigen reagents from 1976–2007 and (**D**) H5 subtype antigen reagents. (**C**) A dose–response curve is shown for nanobody R2b-E8 which only reacted with one antigen reagent out of the H1N1 reagents from 1976–2007 in competitive ELISA. Black dotted lines indicate 95% confidence intervals; red dotted lines indicate average percentage differences.

**Figure 2 vaccines-10-01473-f002:**
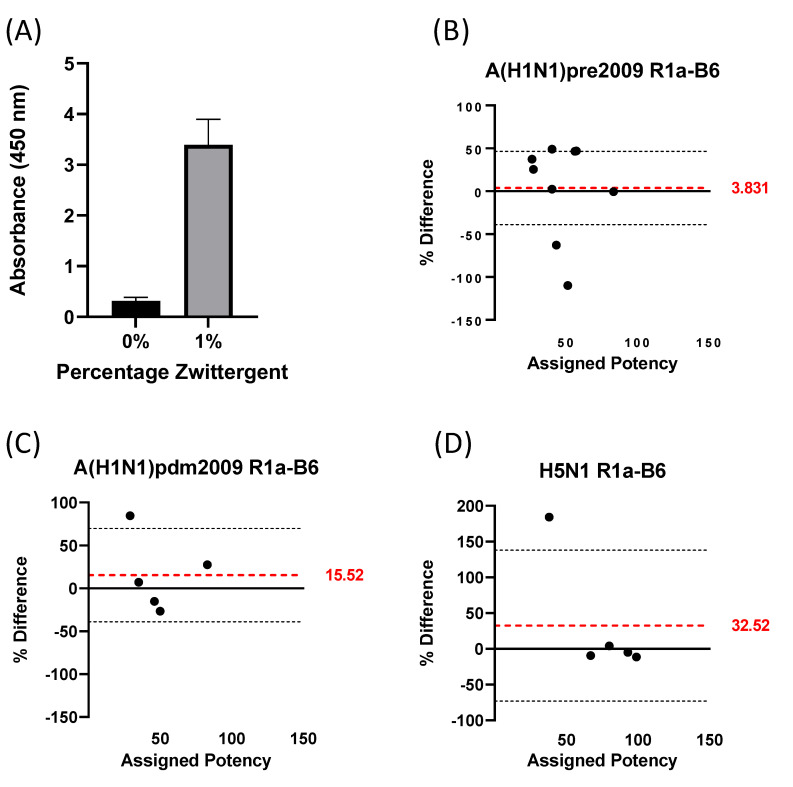
Sandwich ELISA. (**A**) Absorbance signal is shown after pre-treatment of antigen with or without 1% Zwittergent 3–14 detergent. (**B**) Difference plot analysis of results from sandwich ELISA on A(H1N1) antigen reagents from 1976–2007, (**C**) A(H1N1)pdm09 antigen reagents and (**D**) H5 subtype antigen reagents. Assigned potency is in μg/mL; black dotted lines indicate 95% confidence intervals; red dotted lines indicate average percentage differences.

**Figure 3 vaccines-10-01473-f003:**
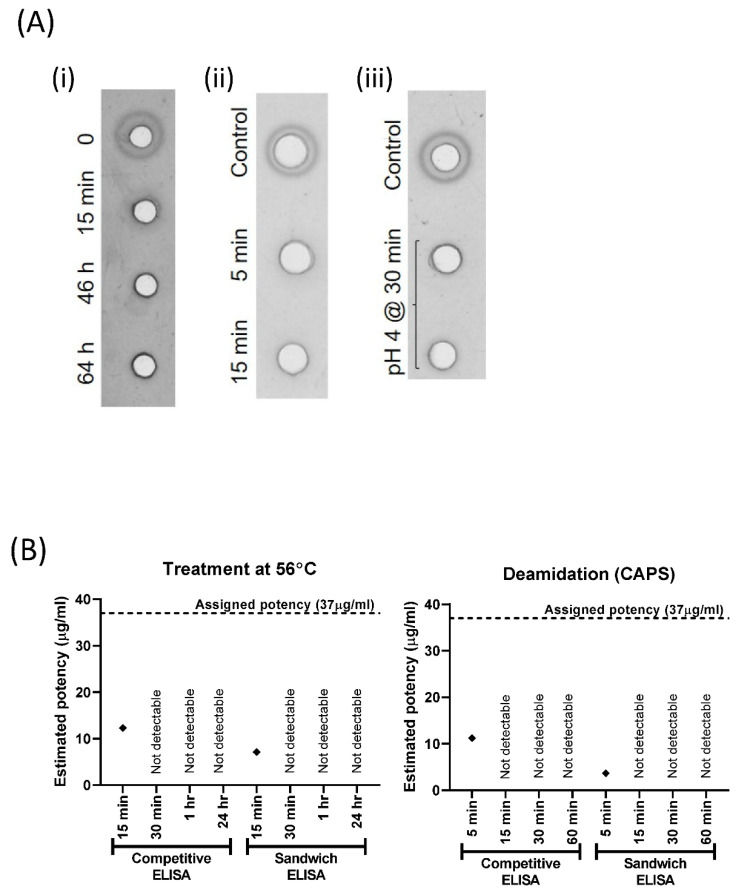
Forced degradation. (**A**) After treatment of antigen reagent (**i**) at 56 °C, (**ii**) by deamidation and (**iii**) by acid treatment, SRD precipitation zones were only present at timepoint 0 (**i**) or in untreated controls (**ii**,**iii**) in SRD assays. (**B**) Estimated potency graphs from ELISA assays after forced degradation at 56 °C or by deamidation.

**Figure 4 vaccines-10-01473-f004:**
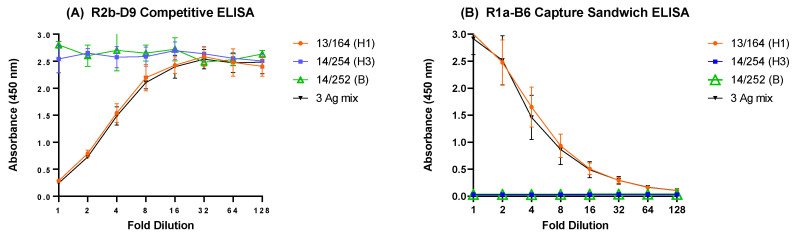
Dose–response of two-fold dilution series of antigen reagents in (**A**) the competitive and (**B**) sandwich ELISA using antigen reagents 13/164 (H1N1)pdm09, 14/254 (H3N2), 14/252 (influenza B) as well as a mix of all three reagents.

**Figure 5 vaccines-10-01473-f005:**
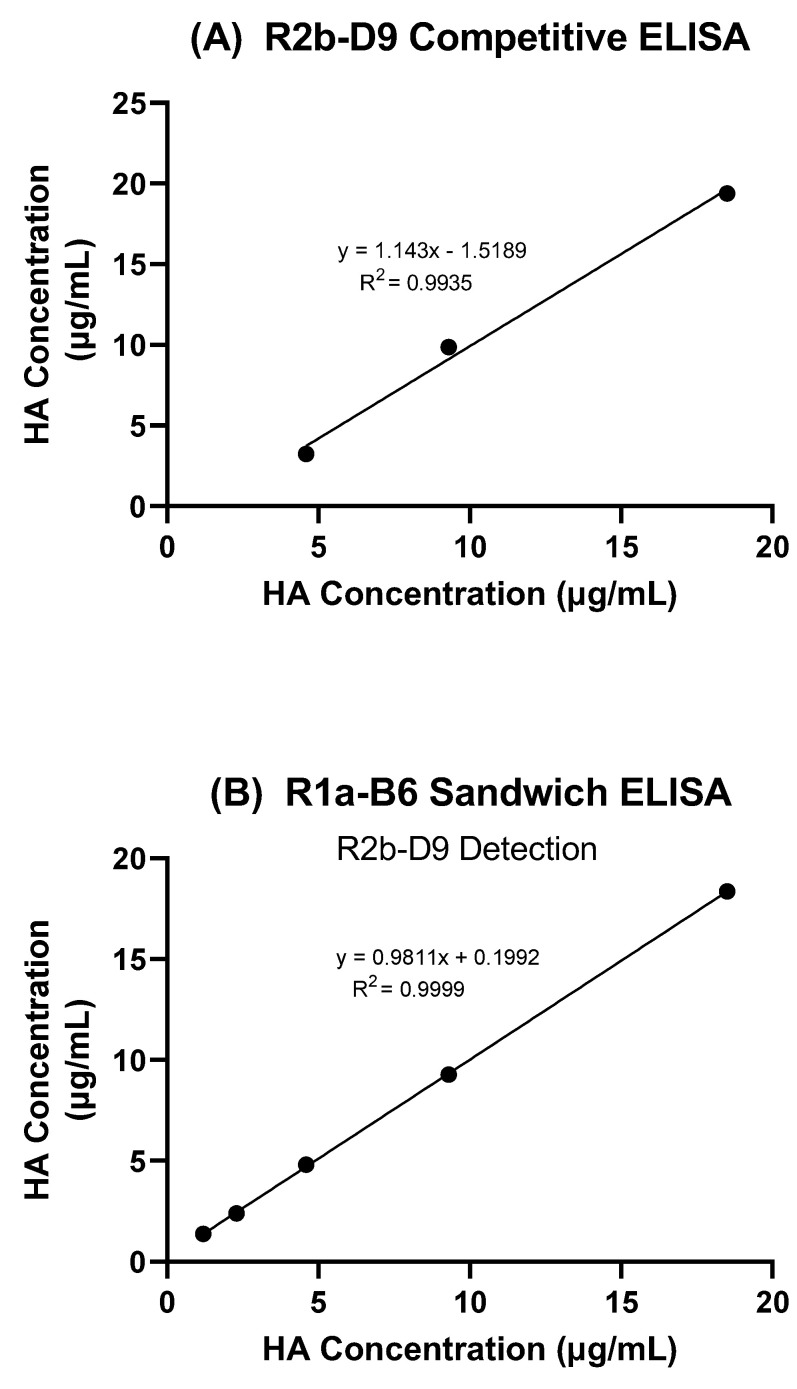
Detection limit and linearity. (**A**) Antigen reagent 09/196 was diluted in 2-fold steps in triplicate, and each dilution was independently analysed by competitive ELISA using R2b-D9 and (**B**) sandwich ELISA using R1a-B6 as capture antibody and R2b-D9 as primary detection antibody. Estimated potency for each dilution is plotted against expected concentration.

**Table 1 vaccines-10-01473-t001:** Antigen reagents used in this study.

Reagent	NIBSC Reference	Virus	Assigned Potency (μg/mL)
**A(H1N1)pdm09 antigens**	13/164	A/California/7/2009 (NYMC X-179A) egg derived antigen	35
	09/174	A/California/7/2009 (NYMC X-179A) cell derived antigen	50
	12/168	A/California/7/2009 (NYMC X-181)	46
	10/258	A/Christchurch/16/2010 (NIB-74)	29
	11/134	A/Brisbane/10/2010 cell derived antigen	83
**A(H1N1)** **antigens** **(1976–2007)**	08/100	A/Brisbane/59/2007 (IVR-148) (H1N1)	83
	07/102	A/Solomon Islands/3/2006 (IVR-145) (H1N1)	57
	06/170	A/New Caledonia/20/99	26
	02/336	A/New Caledonia/20/99	56
	97/760	A/Beijing/262/95	40
	84/538	A/Chile/1/83	27
	79/560	A/Brazil/11/78	40
	79/558	A/USSR/92/77	43
	77/530	A/New Jersey/8/76	51
**A(H5) subtype antigens**	00/552	A/duck/Singapore-Q/F119-3/97 (H5N3)	38
	09/184	A/Vietnam/1194/2004 (H5N1)	67
	07/112	A/turkey/Turkey/1/2005 (H5N1)	80
	07/290	A/Anhui/1/2005 (H5N1)	99
	08/216	A/Cambodia/R0405050/2007 (H5N1)	93
**A**(H3N2) **subtype antigen**	14/254	A/Switzerland/9715293/2013 (NIB-88) (H3N2)	55
**B type antigen**	14/252	B/Phuket/3073/2013	32

**Table 2 vaccines-10-01473-t002:** Correlation between estimated and assigned values.

Nanobody	Pearson’s r		
	A(H1N1)pdm09	H1N1 (1976–2007)	H5
R1a-B6	0.970	0.848	0.516
R2b-D9	0.945	0.716	0.615
R2b-E8	0.925	not applicable	0.581
R1a-A5	0.959	0.712	0.434
R2a-G8	0.927	0.879	0.707

**Table 3 vaccines-10-01473-t003:** Potency estimates of vaccine samples by competitive ELISA.

Vaccine (Manufacturer)	SRD µg HA/mL	Reference Antigens for Potency Estimation											
		Homologous Antigen Reagent (NIBSC 12/168 @ 46 µg/mL)		V1 (Reference @ 23 µg/mL)		V2 (Reference @ 731 µg/mL)		V3 (Reference @ 30 µg/mL)		V4 (Reference @ 33 µg/mL)		V5 (Reference @ 34 µg/mL)	
		Mean Estimated Potency µgHA/mL (CV (%))	Deviation from SRD	Mean Estimated Potency µgHA/mL (CV (%))	Deviation from SRD	Mean Estimated Potency µgHA/mL (CV (%))	Deviation from SRD	Mean Estimated Potency µgHA/mL (CV (%))	Deviation from SRD	Mean Estimated Potency µgHA/mL (CV (%))	Deviation from SRD	Mean Estimated Potency µgHA/mL (CV (%))	Deviation from SRD
V1 Split Trivalent (A)	**23**	46 (0.37)	100%	-	-	22 (12)	−4%	29 (18)	26%	27 (15)	17%	29 (11)	26%
V2 Split Monovalent (H1N1) (B)	**731**	1776 (1.6)	142%	831 (11)	14%	-	-	1040 (5.8)	42%	950 (1.1)	30%	1030 (23)	41%
V3 Split Quadrivalent (B)	**30**	53 (26)	77%	24 (17)	−20%	23 (17)	−23%	-	-	28 (4.1)	−7%	30 (27)	0%
V4 Split trivalent batch 1 (B)	**33**	54 (14)	64%	28 (14)	−15%	27 (12)	−18%	35 (4.8)	6%	-	-	35 (24)	6%
V5 Split trivalent batch 2 (B)	**34**	61 (14)	79%	28 (12)	−18%	27 (22)	−21%	36 (32)	6%	33 (29)	−3%	-	-

**Table 4 vaccines-10-01473-t004:** Potency estimates of vaccine samples by sandwich ELISA.

Vaccine (Manufacturer)	SRD µg HA/mL	Reference Antigens for Potency Estimation											
		Homologous Antigen Reagent (NIBSC 12/168 @ 46 µg/mL)		V1 (Reference @ 23 µg/mL)		V2 (Reference @ 731 µg/mL)		V3 (Reference @ 30 µg/mL)		V4 (Reference @ 33 µg/mL)		V5 (Reference @ 34 µg/mL)	
		Mean Estimated Potency µgHA/mL (CV (%))	Deviation from SRD	Mean Estimated Potency µgHA/mL (CV (%))	Deviation from SRD	Mean Estimated Potency µgHA/mL (CV (%))	Deviation from SRD	Mean Estimated Potency µgHA/mL (CV (%))	Deviation from SRD	Mean Estimated Potency µgHA/mL (CV (%))	Deviation from SRD	Mean Estimated Potency µgHA/mL (CV (%))	Deviation from SRD
V1 Split Trivalent (A)	**23**	42 (14)	83%	-	-	26 (9.8)	13%	25 (14)	9%	25 (14)	9%	27 (12)	17%
V2 Split Monovalent (H1N1) (B)	**731**	1196 (2.7)	64%	660 (10)	−10%	-	-	710 (25)	−3%	710 (25)	−3%	740 (24)	19%
V3 Split Quadrivalent (B)	**30**	50 (8.9)	67%	28 (14)	−7%	32 (22)	7%	-	-	38 (94.9)	27%	32 (98.0)	7%
V4 Split trivalent batch 1 (B)	**33**	63 (14)	91%	36 (17)	9%	41 (21)	24%	38 (5.0)	15%	-	-	40 (13)	21%
V5 Split trivalent batch 2 (B)	**34**	54 (1.6)	59%	30 (12)	−12%	35 (22)	2.9%	32 (7.9)	−6%	32 (7.9)	−6%	-	-

## Data Availability

The data presented in this study are available on request from the corresponding author.

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
