# Peer review of "Development of an ELISA-Based Potency Assay for Inactivated Influenza Vaccines Using Cross-Reactive Nanobodies"

_vaccines, 2022, doi:10.3390/vaccines10091473_

Round 1

Reviewer 1 Report

This is a very useful study reporting quantitative methods potentially not only for seasonal flu vaccines but also pandemic flu vaccine. The authors have run a panel of robust experiments demonstrating the validity of the nanobody-based ELISA assays. Specifically, the assays can detect a very wide range of H1, H5 in addition to H3N2 and IBV. They have compared with the golden-standard assay (SRID) and used stress conditions to ensure the antibodies could detect conformational changes of the HA proteins. The results obtained are strong and do support the potential of this type of ELISA assays for flu vaccine quantifications. Unlike the universal antibodies targeting the linear epitopes of HA & NA, the antibodies are most likely targeting conformational epitopes. It is also impressive that the authors used three conditions, ph, heat and deamidation, in their assay to evaluate the robustness of the ELISA assay.

All in the all, the manuscript was well written with conclusion appropriately drawn based on solid data. Just a couple of minor clarifications are suggested in the revised Ms.

1.       Is there any reason only one strain in H3N2 and IBV was tested, while a lot more was analysed in H1N1 and H5N1.

2.       Could the authors also clarify the selected concentration of Zwittergent (1%), is that to match SRID’s Zwittergent? would different dilutions of Zwittergent affect the assay? Zwittergent at various concentrations have been observed to affect the ELISA assay for vaccine quantifications.

3.       H7N9 might also have pandemic potential. Do you foresee the necessity of testing H7N9 with your assay?

Author Response

Point 1: Is there any reason only one strain in H3N2 and IBV was tested, while a lot more was analysed in H1N1 and H5N1.

The one H3N2 strain was used as a specificity control only. As the nanobodies were not expected to react with Group 2 HAs such as H3-HA and as the H3N2 antigen tested in this study confirmed this, we did not test more H3N2 strains.

Point 2: Could the authors also clarify the selected concentration of Zwittergent (1%), is that to match SRID’s Zwittergent? would different dilutions of Zwittergent affect the assay? Zwittergent at various concentrations have been observed to affect the ELISA assay for vaccine quantifications.

The reviewer is correct: we chose the concentration of Zwittergent to match that used in SRD. SRD is the well-established, gold-standard potency method for influenza HA antigen vaccines, and 1% Zwittergent is used in SRD when testing the various kinds of inactivated vaccines (whole virus, split, subunit) as well as in recombinant HA vaccines; therefore, we wanted to use a concentration known to be suitable for various vaccines. However, we agree with the reviewer that the exact concentration may affect the ELISA assay. We have preliminary evidence that different nanobodies, when used for coating plates, are affected differently by various Zwittergent concentrations. As we only progressed with one nanobody (R1a-B6) in our sandwich ELISA, we stuck to our initial conditions which worked well. If other coating nanobodies were to be used in the future, optimisation of Zwittergent concentration would be advisable.

Point 3: H7N9 might also have pandemic potential. Do you foresee the necessity of testing H7N9 with your assay?

This is a very good question. There may be the need to test for any seasonal virus or zoonotic virus of pandemic potential using alternative potency assays. Our manuscript describes the initial development of a potency assay for Group 1 HAs. H7N9, being a Group 2 HA, would require different nanobodies in similar assay formats. We have described nanobodies for H7N9 that could be used in ELISA assays (DOI: 10.1038/s41598-021-82356-4).

Reviewer 2 Report

The research article is nicely written and observed how ELISA can be used to detect the potency of the vaccine. I have a few points about the article,

1. The author needs to explain the significance of the figures on page 6 and 7.

2. Figure 1, the author needs to number all the graphs properly.

3. Figure 4, the marks on the X-axis should have similar font size.

Author Response

Point 1: The author needs to explain the significance of the figures on page 6 and 7.

Figure 1 (on pages 6 – 8) shows testing of five nanobodies in competitive ELISA against a range of Group 1 antigens (a total of 19 antigens). The results confirm that these nanobodies (with some exceptions for R2b-E8) can be used in a competitive ELISA with H1N1 virus antigens from 1976 to 2010, as well as with H5N1 antigens, confirming their wide cross-reactivity. In addition, the data show that the potency estimates, when using the same standard derived from a pandemic 2009 virus for all antigens, do not always match the expected (assigned) values; together with data in Table 2, these results suggest that use of one standard for all antigens may not be optimal and more matched or re-calibrated standards may be needed for best assay performance. We have added text in lines 195-196 to make the latter point clearer. Also, we have added explicit reference to Figure 1 and Figure 2, where similar findings are reported for the sandwich ELISA, to the Discussion (line 398), to make the link to these data/figures clearer.

Point 2: Figure 1, the author needs to number all the graphs properly.

We have added a third level of labelling, (i) to (v), to identify each graph separately.

Point 3: Figure 4, the marks on the X-axis should have similar font size.

We have double-checked the labelling of the x-axis and found the font size of the numbers below the x-axis to be identical in both graphs. If the reviewer is referring to the marks itself on the x-axis, rather than the numbers below, the marks are identical. However, the blue curve runs roughly along the x-axis in panel B, obscuring the marks somewhat, but the symbols and colour distinguish the blue curve from the x-axis. We’d be happy to modify the figure if we received instructions on what would be best to change.